# Leveraging Distribution Alignment via Stein Path for Cross-Domain Cold-Start Recommendation

**Weiming Liu, Jiajie Su, Chaochao Chen, and Xiaolin Zheng**[*]
Zhejiang University, Hangzhou, China
{21831010,sujiajie,zjuccc,xlzheng}@zju.edu.cn

## Abstract

Cross-Domain Recommendation (CDR) has been popularly studied to utilize different domain knowledge to solve the cold-start problem in recommender systems. In this paper, we focus on the *Cross-Domain Cold-Start Recommendation* (*CDCSR*) problem. That is, how to leverage the information from a source domain, where items are 'warm', to improve the recommendation performance of a target domain, where items are 'cold'. Unfortunately, previous approaches on cold-start and CDR cannot reduce the latent embedding discrepancy across domains efficiently and lead to model degradation. To address this issue, we propose `DisAlign`, a cross-domain recommendation framework for the CDCSR problem, which utilizes both rating and auxiliary representations from the source domain to improve the recommendation performance of the target domain. Specifically, we first propose Stein path alignment for aligning the latent embedding distributions across domains, and then further propose its improved version, i.e., proxy Stein path, which can reduce the operation consumption and improve efficiency. Our empirical study on Douban and Amazon datasets demonstrates that `DisAlign` significantly outperforms the state-of-the-art models under the CDCSR setting.

## 1 Introduction

Data sparsity and cold-start are long-standing problems in recommender systems [15, 19]. With the development of internet techniques, most users always participant in many platforms or domains for different purposes. Therefore, Cross-Domain Recommendation (CDR) has emerged to utilise the relatively richer information from a source domain to improve the recommendation accuracy in a target domain [52, 53]. Most existing CDR models can tackle the data sparsity problem in the target domain by assuming the existence of overlapped users or items with similar tastes or attributions across domains [5].

Instead of focusing on solving the data sparsity problem, we consider cold-start item recommendation under the CDR setting. Specifically, we concentrate on the *Cross-Domain Cold-Start Recommendation* (*CDCSR*) problem, that is, two domains share the same user set but different items, and both domains have auxiliary representations such as item profiles or descriptions. The prime challenge is how to leverage the information from the source domain, where the items are 'warm', to improve the recommendation performance of the target domain, where the items are 'cold'. The CDCSR problem popularly exists in practice, for instance, a movie marketing platform newly launches a book renting service where there is no user-book interaction yet, as shown in Figure 1.

Existing researches on cold-start recommendation and CDR cannot solve the above problem well. On the one hand, existing cold-start recommendation models assume that the distributions of cold items should be consistent with the warm ones as they are homogeneous [54, 18, 12]. On the other hand,

---

[*]Corresponding Author

35th Conference on Neural Information Processing Systems (NeurIPS 2021).

existing CDR models assume that both source and target domains have user-item interactions for learning the mapping functions [25]. Since the cold and warm items are heterogeneous with different latent embedding distributions in practice, and there is no user-item interaction in the target domain, conventional cold-start and CDR models cannot properly suitable to the CDCSR problem.

Similar to the transfer learning task, the key to the CDCSR problem is to reduce the discrepancy between the latent embedding distributions across domains. However, both the warm and cold item representations are scattered and complicated due to the fact that the latent embeddings may represent diverse information. Thus, existing transfer learning based domain adaptation approaches [23, 45, 37] cannot achieve good alignment results, which limits their performances.

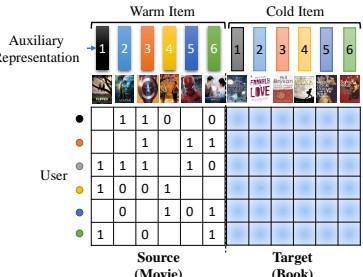

Figure 1: The CDCSR Problem.

To address the aforementioned issue, in this paper, we propose `DisAlign`, a cross-domain recommendation framework for the CDCSR problem. In order to better align the complicated latent embedding distributions and make high quality of rating predictions, we utilize two modules in `DisAlign`, i.e., *rating prediction module* and *embedding distribution alignment module*, as will be shown in Figure 2. The rating prediction module aims to capture user and item collaborative preferences in the source domain, and we propose metric-based contrastive learning for modelling. The goal of distribution alignment module is to properly match the latent embedding distributions across domains, and we propose two techniques for it, i.e., *Stein path alignment* and its improved version called *proxy Stein path alignment*. Specifically, inspired by the particle-based inference algorithm Stein Variational Gradient Descent (SVGD) [21, 13, 20], we first propose Stein path alignment to minimize the domain discrepancy through the particle-moving process, which can take both the source probability and target intra-domain structure into account. Although Stein path can obtain satisfying performance, it has to involve all the target samples during the training process, which is time consuming when data size is large. Thus, we further propose proxy Stein path alignment which only needs to exploit typical samples to represent the target data distribution, and thus can accelerate the operation speed. The comparison and visualization results in experiments will show the reliability and efficiency of `DisAlign`.

We summarize the main contributions of this paper as follows: (1) We propose a novel framework, i.e., `DisAlign`, for the CDCSR problem, which can utilize both rating and auxiliary representations from the source domain to improve the recommendation performance of the target domain. (2) To our best knowledge, this is the first attempt in literature to propose Stein path alignment for aligning the latent embedding distributions across domains, and we also propose its improved version, i.e., proxy Stein path, for higher efficiency. (3) Empirical studies on Douban and Amazon datasets demonstrate that `DisAlign` significantly improves the state-of-the-art models under the CDCSR setting.

## 2 The proposed model

### 2.1 Framework of DisAlign

First, we describe notations. We assume there are two domains, i.e., a source domain $\mathcal{S}$ and a target domain $\mathcal{T}$. We assume both domains $\mathcal{S}$ and $\mathcal{T}$ have $N_U$ users, $\mathcal{S}$ has $N_S$ warm items, and $\mathcal{T}$ has $N_T$ cold items. Let $\boldsymbol{R}^{\mathcal{S}} \in \mathbb{R}^{N_U \times N_S}$ be the warm rating matrix in $\mathcal{S}$ and $\boldsymbol{R}^{\mathcal{T}} \in \mathbb{R}^{N_U \times N_T}$ be the cold rating matrix in $\mathcal{T}$. In CDCSR setting, $\boldsymbol{R}^{\mathcal{T}}$ is absence during training and will be only used for test, since items are cold in $\mathcal{T}$. We also assume that the warm items and the cold items have auxiliary representations $\boldsymbol{X}^W \in \mathbb{R}^{N_S \times Z}$ and $\boldsymbol{X}^V \in \mathbb{R}^{N_T \times Z}$, respectively, with $Z$ denoting

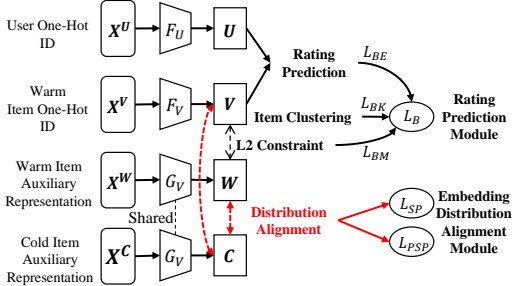

Figure 2: The Framework of `DisAlign`.

the dimension of auxiliary representations. The auxiliary representations usually include useful side-information, e.g., themes, reviews, profiles in a movie domain. Our purpose is to predict the absent $\boldsymbol{R}^{\mathcal{T}}$ in $\mathcal{T}$ by leveraging $\boldsymbol{R}^{\mathcal{S}}$ in $\mathcal{S}$ and the auxiliary representations in both $\mathcal{S}$ and $\mathcal{T}$.

Then, we introduce the overview of our proposed `DisAlign` framework, as is illustrated in Figure 2. `DisAlign` model mainly has two modules, i.e., *rating prediction module* and *embedding distribution alignment module*. To avoid the error superimposition problem [54], the rating prediction module in the source domain mainly provides end-to-end joint training of modelling user and item collaborative embeddings and matching the item collaborative embeddings with item auxiliary embeddings. The embedding distribution alignment module aligns the distributions between the warm and cold items across domains, minimizing the discrepancy between auxiliary latent feature embeddings in the source and target domains sufficiently. We will introduce these two modules in details later.

## 2.2  Rating prediction module

We first introduce the rating prediction module of `DisAlign`. For the $i$-th user and the $j$-th item in the source (warm) domain, we define their corresponding one-hot ID vectors as $\boldsymbol{X}_i^U$ and $\boldsymbol{X}_j^V$, respectively. For the $j$-th warm item, we also define its auxiliary representation as $\boldsymbol{X}_j^W$. The rating prediction module mainly has three purposes, including (1) exploiting user and item collaborative embeddings based on ratings, which is the prime purpose; (2) matching the item collaborative embeddings with item auxiliary embeddings; and (3) obtaining more discriminative item collaborative embeddings using unsupervised clustering method.

First, user and item collaborative embeddings should accurately represent the corresponding rating interactions. We obtain the user and item collaborative embeddings by $F_U(\boldsymbol{X}^U) = \boldsymbol{U} \in \mathbb{R}^{N \times D}$ and $F_V(\boldsymbol{X}^V) = \boldsymbol{V} \in \mathbb{R}^{N \times D}$, respectively. Here, $F_U$ and $F_V$ denote the user and item encoding networks respectively, $N$ is batch size, and $D$ is the dimension of collaborative embeddings. After that, we use pairwise ranking loss $L_{BE}$ based on metric-based contrastive learning [10, 14]:

$$\min L_{BE} = - \sum_{(\boldsymbol{U}_i, \boldsymbol{V}_j, \boldsymbol{V}_k) \in \mathcal{D}} \log \frac{\exp\langle \boldsymbol{U}_i, \boldsymbol{V}_j \rangle}{\exp\langle \boldsymbol{U}_i, \boldsymbol{V}_j \rangle + \exp\langle \boldsymbol{U}_i, \boldsymbol{V}_k \rangle}, \tag{1}$$

where $\mathcal{D} := \{(\boldsymbol{U}_i, \boldsymbol{V}_j, \boldsymbol{V}_k) | R_{ij}^{\mathcal{S}} > R_{ik}^{\mathcal{S}}\}$ denotes the original preference pairs [32], and $\langle \cdot, \cdot \rangle$ denotes the inner product. The loss function $L_{BE}$ can pull the positive items close and push the negative items away for a certain user according to his/her preference.

Second, a item's collaborative embedding should be similar to its auxiliary embeddings to avoid the error superimposition problem [54]. We utilize a network $G_V$ to translate the auxiliary representations into auxiliary embeddings as $G_V(\boldsymbol{X}^W) = \boldsymbol{W} \in \mathbb{R}^{N \times D}$. After it, the item embedding matching loss is given by $L_{BM} = ||\boldsymbol{W} - \boldsymbol{V}||_2^2$.

Third, similar item collaborative embeddings should be clustered in order to obtain more discriminative latent features. We adopt deep unsupervised K-Means clustering approach [24, 48] for this, and the corresponding loss is $\min_{\boldsymbol{F}\boldsymbol{F}^T = \boldsymbol{I}} L_{BK} = \left[ \text{Tr}(\boldsymbol{V}\boldsymbol{V}^T) - \text{Tr}(\boldsymbol{F}\boldsymbol{V}\boldsymbol{V}^T\boldsymbol{F}^T) \right]$, where $\boldsymbol{F} \in \mathbb{R}^{K \times N}$ is the cluster indicator matrix and $K$ denotes the cluster number.

In summary, the loss of the rating prediction module is a combination of the three losses, that is:

$$\min_{\boldsymbol{F}\boldsymbol{F}^T = \boldsymbol{I}} L_B = L_{BE} + \eta L_{BM} + \zeta L_{BK}, \tag{2}$$

where $\zeta$ and $\eta$ represent the balance hyper-parameters. The optimization procedure is given as below: (1) Fixing the other variables except $\boldsymbol{F}$, we update $\boldsymbol{F}$ through $\min_{\boldsymbol{F}\boldsymbol{F}^T = \boldsymbol{I}} \text{Tr}(\boldsymbol{F}\boldsymbol{V}\boldsymbol{V}^T\boldsymbol{F}^T)$ with singular value decomposition algorithm; (2) Fixing $\boldsymbol{F}$, we update other variables through gradient descent methods for several iterations then go back to step (1) until it convergences. For the sake of stability, in practice, we update $\boldsymbol{F}$ every 15 iterations.

## 2.3  Embedding distribution alignment module

### 2.3.1  Overview

We then introduce the embedding distribution alignment module of `DisAlign`. We use $G_V(\boldsymbol{X}_j^C) = \boldsymbol{C} \in \mathbb{R}^{N \times D}$ to denote the auxiliary embeddings of the cold items in the target domain. Specifically, $G_V(\cdot)$ is a two-stream siamese network with shared weights for encoding both warm item auxiliary representation $\boldsymbol{X}^W$ and cold item auxiliary representation $\boldsymbol{X}^C$. We denote $p_W$ and $p_C$ as the warm and cold item auxiliary embedding probability distributions, respectively, and denote $p_V$ as the warm

item collaborative embedding probability distribution. In CDCSR setting, $p_W \neq p_C$ and $p_V \neq p_C$, because the embeddings generated from the source (warm) domain and the target (cold) domain are heterogeneous, which leads to the *domain discrepancy* problem. Let us consider a case where the source domain has user-book interactions while the target domain has user-movie interactions. Although books and movies share some similar characteristics, the auxiliary representations of the Book domain usually include authors and writing styles, while the auxiliary representations of the Movie domain include directors and actors, which brings discrepancy. Without embedding distribution alignment, a recommender system may recommend horror books instead of history books to a user who likes history movies rather than horror movies due to domain discrepancy, as illustrated in the left of Figure 3. After alignment, the history movies/books and horror movies/books are aligned, as is shown in the right of Figure 3, and thus the recommender system can provide more reliable results. In order to reduce the distribution discrepancy between the source and target domains, we introduce two approaches, i.e., Stein path alignment and Proxy stein path alignment.

### 2.3.2 Stein path alignment

As mentioned in Section 1, since the latent embeddings from two domains in CDCSR are scattered and complicated due to the fact that they may represent diverse information, previous domain adaptation methods cannot be effectively utilized to solve the distribution discrepancy problem. Therefore, we propose a new distribution alignment approach, named *Stein path alignment*. Stein path alignment can prompt the target samples move to the source domain through proper *paths*, according to the target

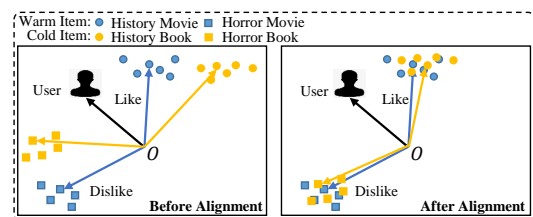

Figure 3: Demonstration of the necessity of embedding distribution alignment.

intra-domain structure and the probability distribution of the source domain. Stein path alignment relies on Stein Variational Gradient Descent (SVGD) [21, 42], a variational inference method that starts from a set of initial particles and iteratively updates them with an approximate steepest direction, whose main iteration process is:

$$\boldsymbol{z}_{i,l} = \boldsymbol{z}_{i,l-1} + \epsilon\phi_p(\boldsymbol{z}_{i,l-1}), \ \phi_p(\boldsymbol{z}) = \frac{1}{N}\sum_{j=1}^{N}[k(\boldsymbol{z}, \boldsymbol{z}_j)\nabla_{\boldsymbol{z}}\log p(\boldsymbol{z}) + \nabla_{\boldsymbol{z}}k(\boldsymbol{z}, \boldsymbol{z}_j)], \qquad (3)$$

where $\boldsymbol{z}_{i,l}$ denotes the $i$-th original target sample at the $l$-th iteration, $p(\boldsymbol{z})$ denotes the source probability distribution, $\epsilon$ denotes the step size, and $k(\boldsymbol{x}, \boldsymbol{y}) = \exp\left(-(\|\boldsymbol{x} - \boldsymbol{y}\|_2^2)/\sigma^2\right)$ is the Gaussian kernel function with $\sigma$ denoting the bandwidth. Existing researches [21, 7, 2] have proved that mean field theory can guarantee the rigorous theoretical convergence of SVGD, that is, the gradient dynamics at particle level will approach to zero: $\lim_{t\to+\infty}(1/N \times \sum_{i=0}^{N}\phi_p(\boldsymbol{z}_{i,t})) \to 0$.

**Stein path distance.** We denote $\boldsymbol{z}_{i,t}$ as the Stein mirror point of $\boldsymbol{z}_{i,0}$ when SVGD convergences at the $t$-th iteration. We propose *Stein path distance* as below:

$$\mathcal{P}_{\mathcal{T}\to\mathcal{S}}(\boldsymbol{Z}) := \frac{1}{N}\sum_{i=0}^{N}\|\boldsymbol{z}_{i,t} - \boldsymbol{z}_{i,0}\|_2^2 = \frac{1}{N}\sum_{i=0}^{N}\|\boldsymbol{z}_{i,t-1} + \epsilon\phi_{\mathcal{S}}(\boldsymbol{z}_{i,t-1}) - \boldsymbol{z}_{i,0}\|_2^2. \qquad (4)$$

Stein path distance quantifies the discrepancy between the source domain $\mathcal{S}$ and the target domain $\mathcal{T}$ by taking the average length of all paths from $\mathcal{T}$ to $\mathcal{S}$ through the $t$-th iteration. Stein path considers the source probability distribution and intra-domain structures, and thus can avoid negative transfer arisen from coarsely pairwise matching by traditional methods. Meanwhile Stein path is also explainable. Let $\boldsymbol{z}_{i,0}$ denote the auxiliary embedding of $i$-th book in the Book domain, $\boldsymbol{z}_{i,t}$ could be taken as a similar movie in the Movie domain, e.g., the movie is based on the story of the book. The calculation of Stein path distance mainly has three steps. First, adopting kernel density estimation [30, 29, 39] with radial basis function kernel to estimate the probabilities of $\boldsymbol{W}$ and $\boldsymbol{V}$. Second, finding the Stein mirror point of the cold item auxiliary embeddings through SVGD by Equation (3). Third, calculating the Stein path distance using Equation (4). The calculation details will be given in Appendix A.1.

**Stein path loss.** In summary, the better the source and target domains are aligned, the smaller the Stein path distance. Therefore, we innovatively propose *Stein path loss* to align the cold item auxiliary

embedding $C$ with warm item collaborative preference $V$ and auxiliary embedding $W$ as below:

$$\min L_{SP} = \mathcal{P}_{C \to W}(C) + \mathcal{P}_{C \to V}(C) + ||\mathcal{P}_{C \to W}(C) - \mathcal{P}_{C \to V}(C)||_2^2, \quad (5)$$

where the first two terms denote the Stein path distances from $C$ to $W$ and $C$ to $V$, respectively, and the third term reinforces that these two distances should be similar.

### 2.3.3 Proxy Stein path alignment

Although Stein path alignment achieves satisfying performance, it has scalability problem when facing large dataset. Because all the cold items in each batch need to be used for calculating Stein path distance. Therefore, it is urgent to reduce the computation cost to accelerate the optimization process. To do this, we propose *proxy Stein path approach* which only needs to choose the most typical cold item proxies to represent the global properties in order to speed up the alignment process through SVGD. We now describe the main steps for finding the proxies and the optimization procedure.

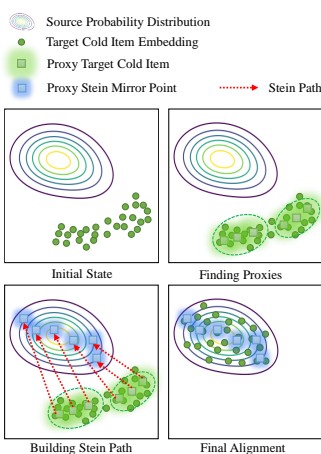

Figure 4: The main procedures of proxy Stein path alignment.

**Multiple-proxies algorithm.** We first introduce a highly efficient *multiple-proxies algorithm*, which aims to find typical proxy samples in the target domain. Suppose there exists $M \in \mathbb{R}^{H \times D}$ typical proxies for the cold item auxiliary embeddings $C$, where $H$ denotes the number of proxies. Let $\Psi \in \mathbb{R}^{N \times H}$ be the similarity matrix between the cold items in the target domain and the proxies. Inspired by [1, 28], we formulate the multiple-proxies optimization problem as

$$\min_{M, \psi_i \mathbf{1}=1, \psi_{ij} \geq 0} \sum_{i=1}^{N} \sum_{j=1}^{H} \psi_{ij} ||c_i - m_j||_2^2 + \alpha \sum_{i=1}^{N} \sum_{j=1}^{H} \psi_{ij} \log \psi_{ij}, \quad (6)$$

where $c_i$ denotes the $i$-th cold item auxiliary embedding and $m_j$ denotes the $j$-th corresponding proxy. The entropy norm regularization term $\sum_{i=1}^{N} \sum_{j=1}^{H} \psi_{ij} \log \psi_{ij}$ is set to avoid trivial solution with $\alpha$ denoting the regularization strength. Compared with the square norm $|| \cdot ||_2$, the entropy norm can not only obtain a nonnegative and nonlinearly representational similarity matrix but also reduce the computational cost [1]. In summary, the multiple-proxies optimization algorithm has two main steps, i.e., (1) updating $\Psi$, which has closed-form solution, and (2) updating $M$ as $m_j = \sum_{i=1}^{N} \psi_{ij} c_i / \sum_{i=1}^{N} \psi_{ij}$. The optimization could be done by repeating (1) and (2) until $\Psi$ and $M$ convergence. We will present the optimization details in Appendix A.2.

**Proxy Stein path distance.** After finding the typical proxies $M$ and setting $\Psi$ as a constant, we propose *proxy Stein path distance* according to the original Stein path as below:

$$\mathcal{P}_{\mathcal{T} \to \mathcal{S}}^*(M) = \frac{1}{H} \sum_{i=0}^{H} ||m_{i,t} - m_{i,0}||_2^2 = \frac{1}{H} \sum_{i=0}^{H} ||m_{i,t-1} + \epsilon \phi_{\mathcal{S}}(m_{i,t-1}) - m_{i,0}||_2^2, \quad (7)$$

where $m_{i,t}$ denotes the $i$-th proxy $m_i$ at the $t$-th iteration. Notably, in each batch, proxy Stein path *only* needs to move the number of proxy samples ($H$) in the target domain rather than the total number of samples ($N$). Since $H < N$, proxy Stein path can reduce the time consumption on calculating the Stein path distance.

**Proxy Stein path loss.** Similarly to the Stein path loss, the *proxy Stein path loss* is given by:

$$\min L_{PSP} = \mathcal{P}_{C \to W}^*(M) + \mathcal{P}_{C \to V}^*(M) + ||\mathcal{P}_{C \to W}^*(M) - \mathcal{P}_{C \to V}^*(M)||_2^2. \quad (8)$$

The optimization of proxy Stein path alignment mainly has four steps. The first step is adopting the multiple-proxies algorithm to figure out the typical proxies $M$ in the target domain. The following three steps are similar as Stein path alignment mentioned in the Section 2.3.2, except that we are moving proxies $M$ rather than $C$. We will present the optimization details in Appendix A.3.

**Time complexity analysis.** The time complexity of Stein path alignment is $O(N^3 t_1)$, where $t_1$ is the iteration number. The time complexities of the multiple-proxies algorithm and proxy Stein path

Table 1: Experimental results on Douban and Amazon datasets.

| | (Douban) Movie→Book | | | (Douban) Movie→Music | | | (Douban) Book→Movie | | | (Amazon) Movie→Music | | |
|---|---|---|---|---|---|---|---|---|---|---|---|---|
| | HR | Recall | NDCG | HR | Recall | NDCG | HR | Recall | NDCG | HR | Recall | NDCG |
| DropoutNet | .2866 | .1528 | .0959 | .2893 | .1896 | .1134 | .2448 | .0976 | .0553 | .2591 | .1463 | .0786 |
| LLAE | .2914 | .1744 | .1078 | .3105 | .2039 | .1278 | .2511 | .1104 | .0618 | .2643 | .1504 | .0850 |
| Heater | .2983 | .1816 | .1135 | .3223 | .2104 | .1310 | .2613 | .1263 | .0707 | .2784 | .1631 | .0942 |
| WCF | .3028 | .1920 | .1266 | .3376 | .2177 | .1385 | .2704 | .1385 | .0796 | .2867 | .1744 | .1103 |
| ESAM | .3146 | .2025 | .1304 | .3467 | .2314 | .1491 | .2815 | .1482 | .0886 | .2942 | .1878 | .1186 |
| DARec | .3139 | .2149 | .1356 | .3350 | .2196 | .1389 | .2749 | .1407 | .0824 | .2981 | .1910 | .1229 |
| DisAlign-Base | .2991 | .1846 | .1189 | .3258 | .2108 | .1324 | .2617 | .1276 | .0733 | .2839 | .1676 | .1054 |
| DisAlign-SP(I) | .3375 | .2373 | .1466 | .3650 | .2485 | .1587 | .3023 | .1682 | .0979 | .3102 | .2096 | .1303 |
| DisAlign-SP | **.3428** | **.2411** | **.1508** | .3734 | .2506 | .1603 | .3028 | .1709 | .1058 | .3155 | .2182 | .1379 |
| DisAlign-PSP | .3401 | .2405 | .1482 | **.3795** | **.2528** | **.1623** | **.3102** | **.1711** | **.1076** | **.3281** | **.2276** | **.1395** |

| | (Douban) Book→Music | | | (Douban) Music→Movie | | | (Douban) Music→Book | | | (Amazon) Music→Movie | | |
|---|---|---|---|---|---|---|---|---|---|---|---|---|
| | HR | Recall | NDCG | HR | Recall | NDCG | HR | Recall | NDCG | HR | Recall | NDCG |
| DropoutNet | .2584 | .1215 | .0686 | .2595 | .1310 | .0767 | .2632 | .1196 | .0603 | .2662 | .1743 | .1005 |
| LLAE | .2685 | .1268 | .0710 | .2635 | .1424 | .0819 | .2717 | .1245 | .0658 | .2753 | .1802 | .1093 |
| Heater | .2724 | .1289 | .0742 | .2701 | .1472 | .0833 | .2834 | .1342 | .0733 | .2848 | .1876 | .1115 |
| WCF | .2710 | .1332 | .0761 | .2722 | .1530 | .0864 | .2726 | .1295 | .0728 | .2967 | .2112 | .1240 |
| ESAM | .2837 | .1398 | .0803 | .2876 | .1709 | .0935 | .2868 | .1486 | .0847 | .3273 | .2204 | .1415 |
| DARec | .2866 | .1410 | .0839 | .2918 | .1683 | .0916 | .2917 | .1409 | .0811 | .3313 | .2293 | .1476 |
| DisAlign-Base | .2712 | .1303 | .0745 | .2684 | .1490 | .0858 | .2746 | .1305 | .0697 | .2913 | .1928 | .1181 |
| DisAlign-SP(I) | .2946 | .1557 | .0881 | .3082 | .1932 | .1107 | .2986 | .1623 | .0957 | .3362 | .2414 | .1520 |
| DisAlign-SP | .2983 | .1581 | .0905 | .3107 | .1948 | .1156 | **.3073** | **.1692** | **.1015** | .3428 | .2505 | .1609 |
| DisAlign-PSP | **.3018** | **.1593** | **.0924** | **.3121** | **.1990** | **.1194** | .3005 | .1644 | .0988 | **.3485** | **.2542** | **.1644** |

alignment are $O(NHt_2)$ and $O(H^2Nt_1)$, respectively, where $t_2$ is the inner-loop iteration number. Therefore, the total time complexity of proxy Stein path alignment is $O(NHt_2) + O(H^2Nt_1) = O(NHt_2 + H^2Nt_1)$. Since $H < N$, proxy Stein path alignment is much cheaper than Stein path alignment. Empirically, we set $H^* = \lceil \frac{N}{2} \rceil$.

## 2.4 Putting together

The total loss of `DisAlign` could be obtained by combining the losses of the rating prediction module and the embedding distribution alignment module. That is, the losses of `DisAlign` with Stein Path (`DisAlign`-**SP**) and Proxy Stein Path (`DisAlign`-**PSP**) are:

$$\min \mathcal{L}_{\texttt{DisAlign-SP}} = L_B + \lambda_{SP} L_{SP}, \ \min \mathcal{L}_{\texttt{DisAlign-PSP}} = L_B + \lambda_{PSP} L_{PSP}, \qquad (9)$$

where $\lambda_{SP}$ and $\lambda_{PSP}$ are hyper-parameters to balance the two types of losses. In testing phase, one can predict the missing rating in the target domain by taking the inner product of user embeddings $U$ and cold item embeddings $C$.

## 3 Experiments

### 3.1 Experimental setup

**Datasets and Tasks.** We conduct extensive experiments on two popurarly used real-world datasets, i.e., *Douban* and *Amazon*. First, the **Douban** dataset [50, 51] has three domains, i.e., Book, Music, and Movie, which contains ratings, reviews, tags, and item details. There are six CDCSR tasks on Douban by randomly choosing two domains as source domain and target domain respectively. Second, the **Amazon** dataset [49, 27] has two domains, i.e., Movies and TV (Movie), and CDs and Vinyl (Music). There are two tasks on Amazon, i.e., Amazon Movie → Amazon Music and Amazon Music → Amazon Movie. For both datasets, we binarize the ratings to 1 and 0. Specifically, we take the ratings higher or equal to 4 as positive and others as negative. We also filter the users and items with less than 5 interactions, following existing research [46, 51]. We list the detailed information on these datasets and tasks in Section B.1 of the supplementary material.

**Baselines.** We compare our proposed `DisAlign` with the following state-of-the-art cold-start and CDR models. (1) **DropoutNet** [41] inputs both auxiliary representations and collaborative filtering representations and randomly dropouts pre-trained collaborative filtering representations for training.

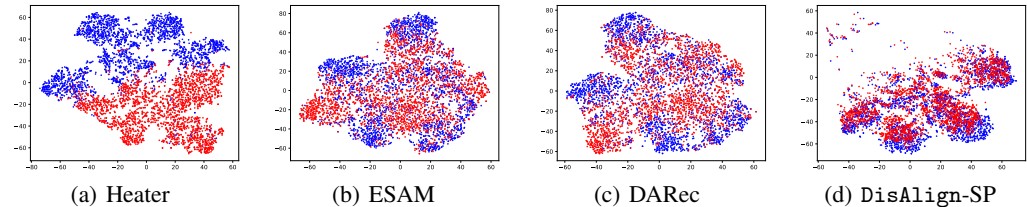

| (a) Heater | (b) ESAM | (c) DARec | (d) DisAlign-SP |

Figure 5: The t-SNE visualization of **Douban Movie**→**Douban Book**, where **Douban Movie** is the source domain with red dots and **Douban Book** is the target domain with blue dots.

(2) **LLAE** [18] introduces semantic auto-encoder using the idea of zero-shot learning to solve the cold-start recommendation problem. (3) **Heater** [54] is the latest cold-start recommendation model which combines separate-training and joint-training framework to overcome the error superimposition issue. (4) **WCF** [26] is the first attempt to apply Wasserstein distance optimal transport for item cold-start recommendation. (5) **ESAM** [4] adopts attribute correlation alignment to improve long-tail recommendation performance by suppressing inconsistent distribution between displayed and non-displayed items. (6) **DARec** [46] adopts adversarial training strategy to extract and transfer knowledge patterns for shared users across domains and achieves the state-of-the-art performance in CDR. For **DropoutNet**, **LLAE**, **Heater**, and **WCF**, we use the same setting as reported in their original papers. For **DARec** and **ESAM**, since they cannot be directly applied to cold-start tasks, we adopt the same rating prediction module as DisAlign. Note that, for a fair comparison, all the models use the same types of data and pre-processing methods during experiments.

**Implemented details.** We provide the implemented details of our proposed model and baselines. The auxiliary representations for $\boldsymbol{X}^W$ and $\boldsymbol{X}^C$ across domains include genres, themes, reviews, item profiles, etc. We split auxiliary representations into each word and adopt directional skip-gram [33] on Douban for Chinese words and apply Glove [31] on Amazon for English words to obtain the average feature representations with dimension $Z = 200$. We use all the user-item rating interactions in the source domain, and all the items auxiliary representations in both the source domain and the target domain for training the model, following standard evaluation for unsupervised adaptation [22, 11]. For all the experiments, we perform five random experiments and report the average results. We choose Adam [16] as optimizer, and adopt Hit Rate@20 (HR@20), Recall@20, and NDCG@20 [43] as the ranking evaluation metrics.

**Hyper-parameter settings.** We set batch size $N = 256$ for both the source and target domains. The latent embedding dimension is set to $D = 128$. For the *rating prediction module*, we set the balance hyper-parameters as $\eta = 0.01$ and $\zeta = 0.01$, and number of cluster $K = 5$ for item unsupervised clustering. For the *stein path alignment module*, we set the moving step size as $\epsilon = 0.01$ and the kernel bandwidth as $\sigma = 0.5$. For the *proxy stein path alignment module*, we set $\alpha = 0.1$ and $H = 64$ according to Section 2.3.3. Finally, for the balance parameters, $\lambda_{SP}$ and $\lambda_{PSP}$ are first selected according to accuracy on **Douban Movie** → **Douban Book** and then fixed as the best values, i.e., $\lambda_{SP} = \lambda_{PSP} = 0.5$. Although there are many hyper-parameters, we first optimize the hyper-parameters of the rating prediction module, and then optimize the other hyper-parameters.

### 3.2 Recommendation performance

**Results and discussion.** The comparison results on Douban and Amazon datasets are shown in Table 1. From it, we can find that (1) Although **Heater** can get better results on conventional cold-start problem, it cannot achieve satisfying solutions on CDCSR problem since it cannot reduce the discrepancy across domains. (2) **WCF** obtains better performance than **Heater** in some tasks, but optimal transport with Wasserstein distance is easily affected by noisy samples, resulting in the over-adaptation errors in boundaries and limiting the transportation results. (3) **ESAM** and **DARec** provide correlated-attribution alignment and adversarial training to match source and target domains, while such coarsely matching methods lead to limited prediction enhancement. (4) DisAlign-**SP** or DisAlign-**PSP** consistently achieves the best performance, which proves that Stein path alignment strategy can significantly improve the prediction accuracy. (5) DisAlign-**PSP** outperforms DisAlign-**SP** on several tasks, e.g., Music and Movie domains on both datasets, which demostrates

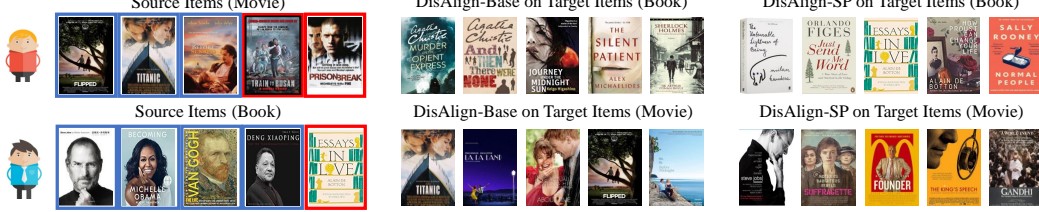

Figure 7: Case study on the recommendation task of **Douban Movie→Douban Book** and **Douban Book→Douban Movie**. The left part are the user preferences in the source domain. The middle and right parts are the recommendation results of `DisAlign`-Base and `DisAlign`-SP, respectively.

that typical proxies can filter out the outliers and improve model robustness. Besides, we further investigate the time consumption of each model on different tasks, and report the results in Figure 6. From it, we find that `DisAlign`-**SP** is the slowest, because it has to transport the whole batchsize of samples from the target domain to the source domain. In contrast, `DisAlign`-**PSP** is much faster than `DisAlign`-**SP**, and also faster than **WCF** and **DARec**, since it only needs to transport typical target proxies.

**Visualization.** To show the feature transferability, we visualize the t-SNE embeddings [17] of the source item auxiliary embeddings ($W$) and the target item auxiliary embeddings ($C$). The results of Douban Movie→Douban Book are shown in Figure 5. From it, we can see that (1) **Heater** does not has the ability to bridge the gap across different domains, and thus the embeddings are separated in source and target domains, as shown in Figure 5(a); (2) **ESAM** and **DARec** have the tendency to draw the source and target embeddings closer, while they still have a certain distance, as shown in Figure 5(b) and Figure 5(c). This indicates that they can only align

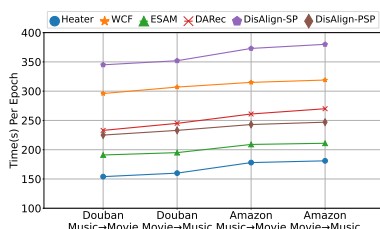

Figure 6: Comparison of running time on four tasks.

the marginal probability distribution; (3) `DisAlign`-**SP** in Figure 5(d) depicts that the embeddings trained through Stein path alignment achieves more closer gap between the source and target domains. The visualization on Amazon dataset shows similar result, and we present it in Section B.2 of the supplementary material.

### 3.3 Analysis

**Ablation study.** To study how does each module of `DisAlign` contribute on the final performance, we compare `DisAlign` with its several variants, including `DisAlign`-**Base** and `DisAlign`-**SP(I)**. (1) `DisAlign`-Base only consists of the rating prediction module with collaborative embeddings clustering. (2) `DisAlign`-SP(I) only aligns the warm item auxiliary embedding $W$ and the cold item auxiliary embedding $C$. The comparison results are shown in Table 1. From it, we can observe that (1) `DisAlign`-Base without the Stein path distribution alignment module cannot transfer knowledge from the source domain to the target domain, resulting in poor performance, (2) `DisAlign`-SP(I) achieves better performance than `DisAlign`-Base, where we only align the distributions between $W$ and $C$, and (3) By extra aligning $C$ with $V$, `DisAlign`-SP can further promote the performance of `DisAlign`-SP(I). Overall, the above ablation study demonstrates that our proposed embedding distribution alignment module is effective in solving the CDCSR problem.

**Case study.** In order to illustrate the domain discrepancy problem mentioned in the Section 2.3 (Figure 3), we visualize the cases on **Douban Movie→Douban Book** and **Douban Book→Douban Movie**. Figure 7 shows the recommendation results. The left part is the user-item interactions in the source domain where the blue and red frames indicate users like or dislike the items respectively. The middle part denotes the top-5 recommendation results for the corresponding users based on `DisAlign`-Base, where we can see that these users will probably dislike the recommended items in the target domain due to the lack of embedding distribution alignment. After applying Stein path alignment on the right part, the recommender system can effectively improve the results. The results indirectly demonstrate that Stein path can properly translate the items across different domains according to the latent probability distribution.

**Parameter sensitivity.** We finally study the effects of hyper-parameters on model performance. First, we vary $\lambda_{SP}$ and $\lambda_{PSP}$ in $\{0.1, 0.3, 0.5, 0.7, 1, 3, 5, 10\}$ and report the results in Section B.3 of the supplementary materials. From it, we see that, the embedding distribution alignment module cannot play a central role in the training process when $\lambda_{SP}, \lambda_{PSP} \to 0$, bringing the discrepancy between the source and target domains. When $\lambda_{SP}$ and $\lambda_{PSP}$ become too large, the embedding distribution alignment module will suppress the rating prediction module, which also decreases the recommendation results. The above results indicate that choosing the proper hyper-parameters to balance the embedding distribution alignment loss and rating prediction loss can effectively improve the performance of `DisAlign`. Then, we study the effect of *embedding dimension* ($D$) on `DisAlign`, and report the results in Section B.4 of the supplementary materials. We find that the recommendation accuracy of `DisAlign`-SP and `DisAlign`-PSP increase with $D$, indicating that larger embedding can represent user and item preferences more precisely.

## 4   Related work

**Cold-start recommendation.** Existing approaches on this are mainly of two types, i.e., separate-training and joint-training. The former models, e.g., LinMap [9] and DeepMusic [40], separate the learning of the collaborative filtering embeddings and auxiliary embeddings, and thus always lead to error superimposition [54]. The later jointly minimizes the recommendation error on the user-item interaction and transformation function, including DropoutNet [41], LLAE [18], and Heater [54]. Among them, Heater [54] is the state-of-the-art model which integrates randomized training mechanism with mixture-of-experts to provide better performance. However, the above cold-start approaches cannot adjust to the CDCSR problem due to the discrepancy across different domains.

**Cross domain recommendation (CDR).** According to [52], existing CDR models have three main types, i.e., *transfer-based* methods, *clustered-based* methods, and *multitask-based* methods. Transfer-based methods [49, 25] learn a linear or nonlinear mapping function across domains. Some recent method [46] even adopts adversarial learning strategy to obtain more reliable knowledge across domains with shared users. Clustered-based methods [44] adopt co-clustering approach to learn cross-domain comprehensive embeddings by collectively leveraging single-domain and cross-domain information within a unified framework. Multi-task-based methods [15, 50, 51, 53] enable dual knowledge transfer across domains by introducing shared connection modules in neural networks. Nevertheless, conventional CDR approaches cannot solve the CSCDR problem where user-item interaction data in absent in the target domain.

**Domain adaptation.** Existing works on this are mainly of three types, i.e., *discrepancy-based* methods, *adversarial-based* methods, and *sample-based matching* methods [55, 36]. Discrepancy-based methods learn the domain-invariant embeddings by the adaptation layer for moment matching, e.g., Maximum Mean Discrepancy (MMD) [3], Correlation Alignment (CORAL) [34, 35], and Center Moment Discrepancy (CMD) [47]. Recently, ESAM [4] extends CORAL by utilizing attribution alignment for domain adaptation in long-tail item recommendation. Adversarial-based methods integrate a domain discriminator for adversarial training, e.g., Domain Adversarial Neural Network (DANN) [8] and Adversarial Discriminative Domain Adaptation (ADDA) [38]. Sample-based matching methods are mainly based on the optimal transport [6, 45], which have the ability of encoding class-structure in distributions for minimizing the global transportation cost. However, the above approaches cannot work well in CDCSR setting, because the latent representations are always more scattered, complicated, and diverse.

## 5   Conclusion

In this paper, we propose Distribution Alignment (`DisAlign`), which includes the *rating prediction module* and the *embedding distribution alignment module*, for solving the cross-domain cold-start recommendation problem. We innovatively propose Stein path alignment and proxy Stein path alignment for embedding alignment across domains. We also conducted extensive experiments to demonstrate the superior performance of our `DisAlign` model. In the future, we plan to extend `DisAlign` to multi-domain cold-start recommendation tasks and conduct more comprehensive experiments on new datasets. We will also modify our model to adapt to the situation where there are also user-item interactions in the target domain.

# 6 Acknowledgments

This work was supported in part by the National Key R&D Program of China (No.2018YFB1403001) and sponsored by the CCF-AFSG Research Fund (No.RF20200005).

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
