# Appendix: Leveraging Distribution Alignment via Stein Path for Cross-Domain Cold-Start Recommendation

**Weiming Liu, Jiajie Su, Chaochao Chen, and Xiaolin Zheng**[*]
Zhejiang University, Hangzhou, China
{21831010,sujiajie,zjuccc,xlzheng}@zju.edu.cn

## A   Model

### A.1   Procedure of Stein path distance

We first present the procedures of Stein path distance calculation in Algorithm 1. The calculation of Stein path distance mainly has three steps. First, adopting kernel density estimation [4, 3] with radial basis function kernel to estimate the source probability of $W$ and $V$ (line 1). Second, finding the Stein mirror point of the cold item auxiliary embeddings through SVGD (line 2-line 8). Third, calculating the Stein path distance (line 9).

### A.2   Procedure of multiple-proxies

As mentioned in Section 2.3.3, the multiple-proxies algorithm is given by:

$$\min_{\boldsymbol{M}, \boldsymbol{\psi}_i \mathbf{1}=1, \psi_{ij} \geq 0} \sum_{i=1}^{N} \sum_{j=1}^{H} \psi_{ij} ||\boldsymbol{c}_i - \boldsymbol{m}_j||_2^2 + \alpha \sum_{i=1}^{N} \sum_{j=1}^{H} \psi_{ij} \log \psi_{ij}, \qquad (1)$$

where $\boldsymbol{c}_i$ denotes the $i$-th cold item auxiliary embeddings and $\boldsymbol{m}_j$ denotes the $j$-th corresponding proxy. We now provide the optimization details on the multiple-proxies algorithm. Alternatively updating $\boldsymbol{M}$ and $\boldsymbol{\Psi}$ can solve Equation (1) efficiently.

**Update $\boldsymbol{\Psi}$.** We first fix the variable $\boldsymbol{M}$ and update $\boldsymbol{\Psi}$. By using Lagrangian multiplier to minimize the objective function, we have:

$$\min_{\boldsymbol{\Psi}} L = \sum_{i=1}^{N} \sum_{j=1}^{H} \psi_{ij} ||\boldsymbol{c}_i - \boldsymbol{m}_j||_2^2 + \alpha \sum_{i=1}^{N} \sum_{j=1}^{H} \psi_{ij} \log \psi_{ij} + \sum_{i=1}^{N} \varpi_i \left( \sum_{j=1}^{H} \psi_{ij} - 1 \right). \qquad (2)$$

Taking the differentiation of Equation (2) w.r.t. $\psi_{ij}$ and setting it to 0, we obtain:

$$\frac{\partial L}{\partial \psi_{ij}} = ||\boldsymbol{c}_i - \boldsymbol{m}_j||_2^2 + \alpha(\log \psi_{ij} + 1) + \varpi_i = \Omega_{ij} + \alpha(\log \psi_{ij} + 1) + \varpi_i = 0. \qquad (3)$$

By solving and simplifying Equation (3), we have:

$$\psi_{ij} = \exp\left( -\frac{\alpha + \varpi_i + \Omega_{ij}}{\alpha} \right) = \exp\left( -\frac{\alpha + \varpi_i}{\alpha} \right) \exp\left( -\frac{\Omega_{ij}}{\alpha} \right). \qquad (4)$$

Meanwhile, taking $\sum_{j=1}^{H} \psi_{ij} = 1$ into Equation (4), we have:

$$\sum_{j=1}^{H} \exp\left( -\frac{\alpha + \varpi_i + \Omega_{ij}}{\alpha} \right) = \exp\left( -\frac{\alpha + \varpi_i}{\alpha} \right) \sum_{j=1}^{H} \exp\left( -\frac{\Omega_{ij}}{\alpha} \right) = 1. \qquad (5)$$

---

[*]Corresponding Author

35th Conference on Neural Information Processing Systems (NeurIPS 2021).

**Algorithm 1** The procedure scheme of Stein path distance $(\boldsymbol{X}^{\mathcal{S}}, \boldsymbol{X}^{\mathcal{T}})$

---

**Input:** $T$: training iteration; $N$: batchsize; $D$: latent dimension; $\sigma$: bandwidth in Gaussian Kernel function; $\boldsymbol{X}^{\mathcal{S}} \in \mathbb{R}^{N \times D}$: source samples; $\boldsymbol{X}^{\mathcal{T}} \in \mathbb{R}^{N \times D}$: target samples.

**Procedure**:

1: Estimate $p_{\boldsymbol{X}^{\mathcal{S}}}$ through Kernel Density Estimation;
2: Initialize $\boldsymbol{X}^{\mathcal{T}}_{i,0} = \boldsymbol{X}^{\mathcal{T}}_i$;
3: **for** $l = 1$ to $T$ **do**
4:    **for** $i = 1$ to $N$ **do**
5:       For all $j = 1, 2, \cdots, N$, calculate $k(\boldsymbol{X}^{\mathcal{T}}_{i,l-1}, \boldsymbol{X}^{\mathcal{T}}_{j,l-1}) = \exp\left(-\frac{||\boldsymbol{X}^{\mathcal{T}}_{i,l-1} - \boldsymbol{X}^{\mathcal{T}}_{j,l-1}||_2^2}{\sigma^2}\right)$;

6:       $\boldsymbol{X}^{\mathcal{T}}_{i,l} = \boldsymbol{X}^{\mathcal{T}}_{i,l-1} + \frac{1}{N} \sum\limits_{j=1}^{N} \left[ k(\boldsymbol{X}^{\mathcal{T}}_{i,l-1}, \boldsymbol{X}^{\mathcal{T}}_{j,l-1}) \nabla_{\boldsymbol{X}^{\mathcal{T}}_{i,l-1}} \log p_{\boldsymbol{X}^{\mathcal{S}}}(\boldsymbol{X}^{\mathcal{T}}_{i,l-1}) + \nabla_{\boldsymbol{X}^{\mathcal{T}}_{i,l-1}} k(\boldsymbol{X}^{\mathcal{T}}_{i,l-1}, \boldsymbol{X}^{\mathcal{T}}_{j,l-1}) \right]$;

7:    **end for**
8: **end for**
9: Calculate the Stein Path $\mathcal{P}_{\mathcal{T} \to \mathcal{S}}(\boldsymbol{X}^{\mathcal{T}}) = \frac{1}{N} \sum\limits_{i=1}^{N} ||\boldsymbol{X}^{\mathcal{T}}_{i,t} - \boldsymbol{X}^{\mathcal{T}}_{i,0}||_2^2$;
10: **Return**: $\mathcal{P}_{\mathcal{T} \to \mathcal{S}}(\boldsymbol{X}^{\mathcal{T}})$;

---

That is,

$$\exp\left(-\frac{\alpha + \varpi_i}{\alpha}\right) = \frac{1}{\sum\limits_{j=1}^{H} \exp\left(-\frac{\Omega_{ij}}{\alpha}\right)}. \tag{6}$$

Thus, the final solution of $\psi_{ij}$ is given by:

$$\psi_{ij} = \frac{\exp\left(-\frac{\Omega_{ij}}{\alpha}\right)}{\sum\limits_{k=1}^{H} \exp\left(-\frac{\Omega_{ik}}{\alpha}\right)}. \tag{7}$$

**Update $\boldsymbol{M}$.** After we have updated $\boldsymbol{\Psi}$, we fix it as a constant and update $\boldsymbol{M}$. Thus, Equation (1) becomes

$$\min_{\boldsymbol{M}} \sum_{i=1}^{N} \sum_{j=1}^{H} \psi_{ij} ||\boldsymbol{c}_i - \boldsymbol{m}_j||_2^2. \tag{8}$$

Taking the differentiation of Equation (8) w.r.t. $\boldsymbol{m}_j$ and setting it to 0, we can update $\boldsymbol{M}$ as:

$$\boldsymbol{m}_j = \frac{\sum\limits_{i=1}^{N} \psi_{ij} \boldsymbol{c}_i}{\sum\limits_{i=1}^{N} \psi_{ij}}. \tag{9}$$

We finally summarize the optimization of multiple-proxies in Algorithm 2.

---

**Algorithm 2** The procedure scheme of Multiple-proxies

---

**Input:** $t$: training iteration; $N$: batch size; $D$: latent dimension; $\alpha$: the hyper parameter between the main objective loss and regularization term; $\boldsymbol{C} \in \mathbb{R}^{N \times D}$: cold item auxiliary embedding;

**Procedure**:

1: Random initialize the proxies $\boldsymbol{M}$.
2: **for** $i = 1$ to $t$ **do**
3:    Updating $\boldsymbol{\Psi}$ through Equation.(7)
4:    Updating $\boldsymbol{M}$ through Equation.(9)
5: **end for**
6: **Return**: $\boldsymbol{\Psi}, \boldsymbol{M}$.

---

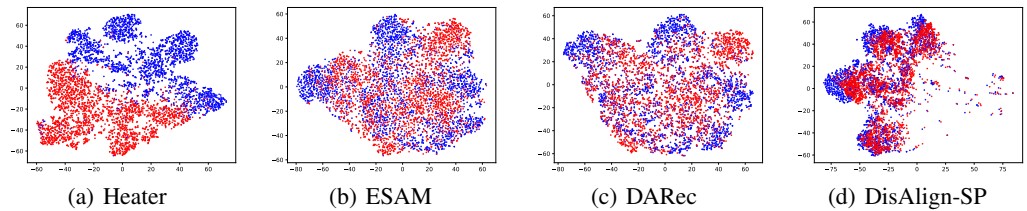

|          |          |          |              |
| :------: | :------: | :------: | :----------: |
| (a) Heater | (b) ESAM | (c) DARec | (d) DisAlign-SP |

Figure 1: The t-SNE visualization of **Amazon Movie→Amazon Music**. **Amazon Movie** is the warm domain with blue dots and **Amazon Music** is the cold-start domain with red dots.

### A.3 Procedure of proxy Stein path loss

As we have presented in Section 2.3.3, the *proxy Stein path distance* is defined as:

$$\mathcal{P}^*_{\mathcal{T}\to\mathcal{S}}(\boldsymbol{M}) = \frac{1}{H}\sum_{i=0}^{H}||\boldsymbol{m}_{i,t}-\boldsymbol{m}_{i,0}||_2^2 = \frac{1}{H}\sum_{i=0}^{H}||\boldsymbol{m}_{i,t-1}+\epsilon\phi_{\mathcal{S}}(\boldsymbol{m}_{i,t-1})-\boldsymbol{m}_{i,0}||_2^2, \quad (10)$$

where $\boldsymbol{m}_{i,t}$ denotes the $i$-th proxy $\boldsymbol{m}_i$ at the $t$-th iteration. Notably, in each batch, proxy Stein path *only* needs to move the number of proxy samples ($H$) in the target domain rather than the total number of samples ($N$). We now briefly demonstrate that the original target samples $\boldsymbol{C}$ can be updated by the typical proxies $\boldsymbol{M}$ through gradient descend. By taking the gradient of $\mathcal{P}^*_{C\to W}(\boldsymbol{M})$ w.r.t. $\boldsymbol{c}_{i,0}$, we have:

$$\frac{\partial\mathcal{P}^*_{C\to W}(\boldsymbol{M})}{\partial\boldsymbol{c}_{i,0}} = \sum_{j=1}^{H}\frac{\partial\mathcal{P}^*_{C\to W}(\boldsymbol{M})}{\partial\boldsymbol{m}_{j,0}}\frac{\partial\boldsymbol{m}_{j,0}}{\partial\boldsymbol{c}_{i,0}} = \sum_{j=1}^{H}\left(-\frac{2}{H}\left(\boldsymbol{m}_{j,t}-\boldsymbol{m}_{j,0}\right)\right)\frac{\psi_{ij}}{\sum_{i=1}^{N}\psi_{ij}}$$

$$= \sum_{j=1}^{H}\left(-\frac{2}{H}\left(\boldsymbol{m}_{j,t}-\frac{\sum_{i=1}^{N}\psi_{ij}\boldsymbol{c}_{i,0}}{\sum_{i=1}^{N}\psi_{ij}}\right)\right)\frac{\psi_{ij}}{\sum_{i=1}^{N}\psi_{ij}}. \quad (11)$$

Obviously, $\boldsymbol{m}_{j,0}$ is the weighted sum of $\boldsymbol{C}$ according to Equation (9), therefore the cold item auxiliary embedding $\boldsymbol{C}$ can be updated through the typical proxies $\boldsymbol{M}$. We present the procedure of proxy Stein path loss in Algorithm 3.

---

**Algorithm 3** The procedure scheme of proxy Stein path loss

---

**Input:** $N$: batchsize; $D$: latent dimension; $\boldsymbol{V}\in\mathbb{R}^{N\times D}$: warm item preference embedding; $\boldsymbol{W}\in\mathbb{R}^{N\times D}$: warm item auxiliary embedding; $\boldsymbol{C}\in\mathbb{R}^{N\times D}$: cold item auxiliary embedding;
**Procedure**:
  1: Finding Multiple Proxies $\boldsymbol{M}$ on $\boldsymbol{C}$ through Algorithm 2.
  2: Calculating the Stein Path distance $\mathcal{P}^*_{C\to W}(\boldsymbol{M})$ through Algorithm 1.
  3: Calculating the Stein Path distance $\mathcal{P}^*_{C\to V}(\boldsymbol{M})$ through Algorithm 1.
  4: **Return:** $L_{PSP} = \mathcal{P}^*_{C\to W}(\boldsymbol{M}) + \mathcal{P}^*_{C\to V}(\boldsymbol{M}) + ||\mathcal{P}^*_{C\to W}(\boldsymbol{M}) - \mathcal{P}^*_{C\to V}(\boldsymbol{M})||_2^2.$

---

## B Experiment

### B.1 Datasets

We conduct extensive experiments on two popularly used real-world datasets, i.e., *Douban* [6, 7] and *Amazon* [5, 2]. The details of Douban and Amazon datasets are shown in Table 1 and Table 2.

### B.2 Visualization

To show feature transferability, we visualize the t-SNE embeddings [1] of the source item auxiliary embeddings ($\boldsymbol{W}$) and the target item auxiliary embeddings ($\boldsymbol{C}$). The results of Amazon

Table 1: Experimental datasets and tasks on Douban and Amazon datasets

| Datasets | Items | Users | Interactions | Density |
|---|---|---|---|---|
| Douban Movie | 34,893 | 2,712 | 1,278,401 | 1.35% |
| Douban Book | 6,777 | 2,110 | 96,041 | 0.67% |
| Douban Music | 5,567 | 1,672 | 69,709 | 0.75% |
| Amazon Movie (Movies and TV) | 12,287 | 27,822 | 779,376 | 0.228% |
| Amazon Music (CDs and Vinyl) | 7,710 | 11,053 | 296,188 | 0.348% |

Table 2: Statistics on different CDCSR tasks

| Source datasets | Target datasets | #Overlap users |
|---|---|---|
| Douban Book | Douban Movie | 2,106 |
| Douban Music | Douban Movie | 1,666 |
| Douban Music | Douban Book | 1,562 |
| Amazon Movie (Movies and TV) | Amazon Music (CDs and Vinyl) | 2,782 |

Table 3: Ablation test on $H$

| (Amazon) Music $\rightarrow$ Movie | $H = \sqrt{N}$ | $H = \frac{1}{4}N$ | $H = \frac{1}{3}N$ | $H = \frac{1}{2}N$ | $H = \frac{2}{3}N$ | $H = \frac{3}{4}N$ | $H = N$ |
|---|---|---|---|---|---|---|---|
| HR | 0.3447 | 0.3461 | 0.3471 | 0.3485 | 0.3480 | 0.3455 | 0.3428 |
| Recall | 0.2518 | 0.2524 | 0.2539 | 0.2542 | 0.2535 | 0.2522 | 0.2505 |
| NDCG | 0.1612 | 0.1624 | 0.1633 | 0.1644 | 0.1639 | 0.1625 | 0.1609 |

Movie$\rightarrow$Amazon Music are shown in Figure 1. From it, we find that the conclusion is similar as Douban Movie$\rightarrow$Douban Book, as we have reported in Section 3.2. That is: (1) **Heater** does not has the ability to bridge the gap across different domains, and thus the embeddings are separated in source and target domains, as shown in Figure 1(a); (2) **ESAM** and **DARec** have the tendency to draw the source and target embeddings closer, while they still have a certain distance, as shown in Figure 1(b) and Figure 1(c). This indicates that they can only align the marginal probability distribution; (3) `DisAlign`-**SP** in Figure 1(d) depicts that the embeddings trained through Stein path alignment achieve more closer gap between source and target domains.

### B.3 Parameter Sensitivity

We also study the effect of hyper-parameters $\lambda_{SP}$ and $\lambda_{PSP}$ on our proposed `DisAlign`-SP and `DisAlign`-PSP. We vary $\lambda_{SP}$ and $\lambda_{PSP}$ in $\{0.1, 0.3, 0.5, 0.7, 1, 3, 10\}$ on two CDCSR tasks, i.e., **Douban Movie$\rightarrow$Douban Book** and **Douban Book$\rightarrow$Douban Movie**. Figure 2 shows the bell-shaped curve, indicating that choosing the proper hyper-parameters to balance the embedding distribution alignment loss and rating prediction loss can effectively improve the model performance. When $\lambda_{SP}, \lambda_{PSP} \rightarrow 0$, the embedding distribution module cannot play a part of role in the training process, causing the discrepancy between the source and target domains. Finding the proper trade-off between the rating prediction loss and embedding distribution alignment loss when $\lambda_{SP} = 0.5$ and $\lambda_{PSP} = 0.5$ can obtain the best performance on both datasets. Moreover, we even conduct the experiments on the $H = \{\sqrt{N}, \frac{N}{4}, \frac{N}{3}, \frac{N}{2}, \frac{2}{3}N, \frac{3}{4}N, N\}$ respectively on **Amazon Music $\rightarrow$ Amazon Movie**. The result has been shown in Table 3. In most cases, when the cold-item embedding space is clustered, the accuracy will gradually increase with the increase of $H$ (e.g., from $H = \sqrt{H}$ to $H = \frac{N}{2}$). However, when $H$ further approaches $N$, the accuracy will decrease since some outliers may cause side-effects. Meanwhile bigger $H$ will cause longer time consumption. In our paper, to achieve a good balance between time complexity and prediction performance, we set $H = [\frac{N}{2}]$. In fact, one can set to a smaller value, e.g., $H = \sqrt{N}$, where the time complexity will decrease to $O(N^2)$. Naturally, this comes with some accuracy loss. However, the prediction accuracy of Proxy Stein path alignment still slightly outperforms Stein path alignment when the cold-item embedding space is clustered.

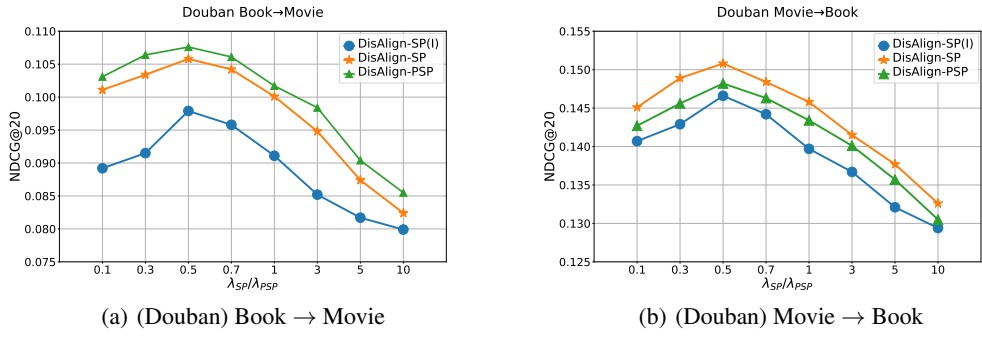

(a) (Douban) Book → Movie

(b) (Douban) Movie → Book

Figure 2: Effect of sensitivity $\lambda_{SP}$ and $\lambda_{PSP}$ on model performance.

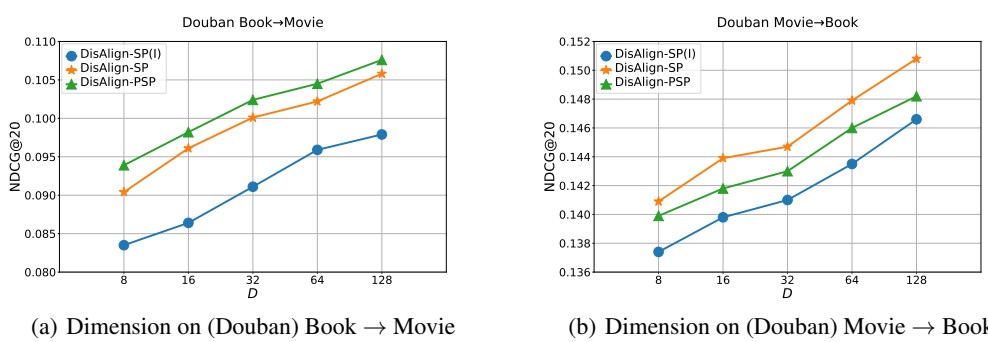

(a) Dimension on (Douban) Book → Movie

(b) Dimension on (Douban) Movie → Book

Figure 3: Effect of embedding dimension $D$ on model performance.

## B.4  Embedding Dimension

We finally analysis the effect of latent embedding dimension $D$ on the performance of our proposed `DisAlign`-SP and `DisAlign`-PSP on two tasks, i.e., **Douban Movie→Douban Book** and **Douban Book→Douban Movie**. The results are shown in the Figure 3, where we range $D$ in $\{8, 16, 32, 64, 128\}$. From it, we can see that, the recommendation accuracy of `DisAlign`-SP and `DisAlign`-PSP increase with $D$, which indicates that a larger embedding can provide a more accurate latent embeddings for both users or items.