# OpenReview forum: "Leveraging Distribution Alignment via Stein Path for Cross-Domain Cold-Start Recommendation"
_NeurIPS.cc/2021/Conference — NeurIPS 2021 Poster_

### Official Review · Reviewer_RKBg · 2021-07-14

**Rating:** 6
**Confidence:** 4

**Summary:**

This paper proposes a framework for cross-domain item cold-start recommendation. The main idea follows the algorithm of Stein Variational Gradient Descent, by iteratively transporting the auxiliary embeddings of cold items in the target domain to match the embedding distribution of warm items in the source domain. The authors further propose to reduce computational time costs by selecting typical proxies of cold items.

**Limitations And Societal Impact:**

Yes.

**Main Review:**

Strength
1. This paper is well-organized and mostly clearly written.
2. It is technically sound to apply SVGD to transport the auxiliary embeddings of cold items to match the embedding distribution of warm items.
3. It is novel to reduce the time cost of SVGD by finding typical proxy samples in the target domain.
4. The experimental results are promising compared with existing methods.
---
Weakness
1. It is not clear to me why the auxiliary features of items in different domains could be encoded with a shared network. For example, an item in the movie domain may have three feature fields: director, actor, category while an item in the book domain may have two different feature fields: author, book name. In this way, it doesn't make sense to use a shared network to encode their features. This may limit the applicability of the proposed method in real-world cross-domain recommendation systems. Besides, the so-called two-stream siamese network is not defined, which should be better demonstrated.
2. In the Proxy Stein path alignment section, it is not shown how the embeddings of cold items are updated. Although I am able to understand the general procedure after reading the supplemental materials, it is still suggested to make it clearer in the paper.
3. It is not shown why Equation (2) has to be solved through alternating optimization. There is no proof of the convergence of the alternating optimization procedure. Also, it is not reasonable to say updating F every 15 iterations could guarantee stability.
4. Similar to 3, there is no proof of the convergence of the alternating optimization of multiple proxies in Equation (6).
5. Although the proposed proxy approach can reduce the time cost of the original SVGD, the time complexity is still $O(N^3)$, since $H^*=[\frac{N}{2}]$. Thus the proposed method may still hardly scale to industrial scenarios. Besides, it would be helpful to see how would the size of proxies affect the model performance.
---
AFTER REBUTTAL: The authors' responses addressed most of my concerns, and there are still some improvements to be made in the paper for better clarification. Thus I would like to increase my rating to above the threshold.

**Time Spent Reviewing:**

6

---

> ### Author Response · Authors · 2021-08-10
> **Response to Review RKBg**
>
> Thanks for your insightful comments. Your valuable comments have greatly enlighted us.
>
> **Q1**: Why the auxiliary feature of items in different domains could be encoded with a shared network? How to adopt the siamese network to encode their features?
>
> **A1**: Indeed, different domains always have different features. For example, in our experiments, the movie domain has the features like director, actor, and category, while the book domain has the features such as author and book name. To handle these features, we filter the stopping words and adopt Word2Vec [1,2] to obtain each word's embedding. After it, we average the embeddings of these words to obtain the initial auxiliary representation for each item. The similar preprocessing method had also been adopted in [3]. In fact, we have described these procedures in line 267--269, and we will add necessary supplements in the camera-ready version. The siamese network is popularly used in recommendation systems [3,4]. We have simply introduced the siamese network in line 132, where we denote $G_V$ as the siamese network with fully connected networks. We will also explain it more clearly in the camera-ready version.
>
> **Q2**: How to update the embeddings of cold items during the training procedure？
>
> **A2**: Due to the space limited, we had to present these parts in the supplementary materials instead of the main paper. In short, the gradient can be backpropagated to the original target samples $N$ through the proxies $H$. We will add these descriptions in the main paper to make it more readable during revision.
>
> **Q3**: Why Equation (2) should be solved through alternating optimization? How about the convergence and stability?
>
> **A3**: The main reason is that $\pmb{F}$ is orthogonal due to the constraint of $\pmb{FF}^T =\pmb{I}$. Although it can be substituted with a soft constraint, it will reduce the performance. The convergence of this optimization has been proven in previous NeurIPS publications [5,6].  We set the update procedure for 15 iterations empirically according to paper [5].
>
> **Q4**: How about the convergence of the Equation (6)?
>
> **A4**: The proof of the convergence for Equation (6) is similar to that for K-Means in Expectation-Maximum algorithm [7]. The first step that optimizes $\pmb{\Phi}$ can be seen as the Expectation step and the second step that optimizes $m$ can be seen as the Maximum step. Both the first part and the second part are convex optimization problems, therefore it is a global optimization problem [7]. We will add the illustrations in the final version to make it more clearer.
>
> **Q5**: How about the details of time complexity and how the typical proxies affect the model performance?
>
> **A5**: $H$ is a tunable hyper-parameter that balances accuracy and efficiency. When $H$ approaches to 0, it will lack the ability of representing the total original data distributions. In most cases, when the cold-item embedding space is clustered, the accuracy will gradually increase with the increase of $H$ (e.g., from $H = \sqrt{N}$ to $H = \frac{1}{2}N$). However, when $H$ further approaches $N$, the accuracy will decrease since some outliers may cause side-effects. Meanwhile bigger $H$ will cause longer time consumption. In our paper, to achieve a good balance between time complexity and prediction performance, we set $H=[\frac{N}{2}]$. In fact, one can set $H$ to a smaller value, e.g., $H = \sqrt{N}$, where the time complexity will decrease to $O(N^{2})$. Naturally, this comes with some accuracy loss. However, the prediction accuracy of Proxy Stein path alignment still slightly outperforms Stein path alignment when the cold-item embedding space is clustered. We provide some experimental results (e.g., Amazon Book -> Amazon Movie) in the following table and are happy to add them in the camera-ready version.
>
> | Amazon Book->Movie | $H = \sqrt{N}$ | $H = \frac{1}{4}N$ | $H = \frac{1}{3}N$ | $H = \frac{1}{2}N$ | $H = \frac{2}{3}N$ | $H = \frac{3}{4}N$ | $H = N$ |
> | ------------------ | ----------- | ------- | ------- | ------- | -------- | -------- | ------ |
> | HR                 | 0.3447      | 0.3461  | 0.3471  | 0.3485  | 0.3480   | 0.3455   | 0.3428 |
> | Recall             | 0.2518      | 0.2524  | 0.2539  | 0.2542  | 0.2535   | 0.2522   | 0.2505 |
> | NDCG               | 0.1612      | 0.1624  | 0.1633  | 0.1644  | 0.1639   | 0.1625   | 0.1609 |
>
> **Ref** :
>
> [1] Yan Song, Shuming Shi, Jing Li, and Haisong Zhang. Directional skip-gram: Explicitly distinguishing left and right context for word embeddings. In Proceedings of the 2018 Conference of the North American Chapter of the Association for Computational Linguistics: Human Language Technologies, Volume 2 (Short Papers), pages 175–180, 2018.
>
> [2] Jeffrey Pennington, Richard Socher, and Christopher Manning. Glove: Global vectors for word representation. volume 14, pages 1532–1543, 01 2014.
>
> [3] Zhihong Chen, Rong Xiao, Chenliang Li, Gangfeng Ye, Haochuan Sun, and Hongbo Deng. Esam: Discriminative domain adaptation with non-displayed items to improve long-tail performance. In SIGIR, pages 579–588, 2020.
>
> [4] Huang, P.-S.; He, X.; Gao, J.; Deng, L.; Acero, A.; and Heck,L. 2013. Learning deep structured semantic models for web search using clickthrough data. In Proceedings of the 22nd ACM international conference on Information & Knowledge Management, 2333–2338.
>
> [5] Qianli Ma, Jiawei Zheng, Sen Li, and Gary W Cottrell. Learning representations for time series clustering. Advances in neural information processing systems, 32:3781–3791, 2019.
>
> [6] Hongyuan Zha, Xiaofeng He, Chris Ding, Ming Gu, and Horst D Simon. Spectral relaxation for k-means clustering. In NIPS, pages 1057–1064, 2001.
>
> [7] Liang Bai and Jiye Liang. Sparse subspace clustering with entropy-norm. In ICML, pages 561–568. PMLR, 2020.

---

> > ### Comment · Reviewer_RKBg · 2021-08-30
> > **A further question about Q1.**
> >
> > Thanks for your clarification. According to your answer, does it mean that the method can only be used for word-based auxiliary features and is not available for ID-based or numerical features? If so, the problem should be clearly defined, since it is not a default setting for cross-domain recommendation.

---

> > > ### Author Response · Authors · 2021-08-30
> > > **Response to Review RKBg**
> > >
> > > Thanks so much for your response. Please allow us to clarify your comments below.
> > >
> > > **Q1**: Is proposed model DisAlign can only be used for word-based auxiliary features and is not available for ID-based or numerical features?
> > >
> > > **A1**: Our proposed DisAlign can be applied for many other kinds of auxiliary features besides word-based ones. We will provide the illustration details below.
> > >
> > > First, we propose a general framework (i.e., DisAlign in Figure 2) to solve the Cross-Domain Cold-Start Recommendation (CDCSR) problem. The proposed model can better align the warm and cold item embeddings to provide better recommendation results across domains.
> > >
> > > Second, word-based auxiliary features are commonly used in classical cold-start models, e.g.,  DropoutNet [1] (NeurIPS 2017) and Heater [2] (SIGIR 2020), to serve as auxiliary information. Specifically, most widely used cold-start datasets CityULike [3] utilizes the information including the title, abstract and article content information which are all the word-semantic information. One can adopt word2vec to embed these useful word-based auxiliary information. Moreover, more and more researchers have shown that the word-based auxiliary features are also highly useful, especially for the Douban and Amazon datasets for cross domain recommendation [4]. Therefore we choose the word-based auxiliary features in our experiments to validate our model performance.
> > >
> > > Third, besides the word-based auxiliary features, our model can be applied to some other features (including numerical features) as well. The only difference is using suitable methods to extract information from different kinds of features. For example, one can extract the source and target item images  through the pre-trained deep aesthetic neural network (ILGNet) [5,6] and use the extracted latent image embeddings as auxiliary features. For numerical features, one can use multi-layer perception or other network structures to extract latent embeddings. Generally speaking, word-based auxiliary features are commonly available and used in the cross-domain recommendation and cold-start recommendation tasks [1,2,3,4,7,8,9,10], including tags, profiles etc. This is because word-based auxiliary features are always more representative than other numerical features in the specific tasks. Therefore, we utilize the word-based auxiliary features in our experiments based on the previous researches [4,7,8,9,10]. We will add these explanations in the camera-ready version.
> > >
> > > As a summary, in this paper, we are focusing on a novel problem in CDR, i.e., the Cross-Domain Cold-Start Recommendation (CDCSR) problem, instead of a traditional CDR problem. That is, how to leverage the information from a source domain, where items are ‘warm', to improve the recommendation performance of a target domain, where items are ‘cold'. Under such a setting, we focus on the innovation of the alignment method between the warm and cold item domains. We believe our empirical study, using word-based auxiliary features, meets the most common CDCSR situations in practice.  One can also choose different preprocessing methods to handle other types of features in practice.
> > >
> > > We hope our response can address your concern on the auxiliary feature. We are also looking forward to your further comments.
> > >
> > > **Ref**:
> > >
> > > [1] Volkovs, Maksims, Guang Wei Yu, and Tomi Poutanen. "DropoutNet: Addressing Cold Start in Recommender Systems." NIPS. 2017.
> > >
> > > [2] Zhu, Ziwei, et al. "Recommendation for new users and new items via randomized training and mixture-of-experts transformation." Proceedings of the 43rd International ACM SIGIR Conference on Research and Development in Information Retrieval. 2020.
> > >
> > > [3] Chong Wang and David M Blei. 2011. Collaborative topic modeling for recommending scientific articles. In Proceedings of the 17th ACM SIGKDD international conference on Knowledge discovery and data mining. ACM, 448–456.
> > >
> > > [4] Feng Zhu, Yan Wang, Chaochao Chen, Guanfeng Liu, and Xiaolin Zheng. A graphical and attentional framework for dual-target cross-domain recommendation. In IJCAI, pages 3001–3008, 2020.
> > >
> > > [5] Liu, Jian, et al. "Exploiting aesthetic preference in deep cross networks for cross-domain recommendation." Proceedings of The Web Conference 2020. 2020.
> > >
> > > [6] Xin Jin, Le Wu, Xiaodong Li, Xiaokun Zhang, Jingying Chi, Siwei Peng, Shiming Ge, Geng Zhao, and Shuying Li. 2019. ILGNet: inception modules with connected local and global features for efficient image aesthetic quality classification using domain adaptation. IET Computer Vision 13, 2 (2019), 206–212.
> > >
> > > [7] Pugoy, Reinald Adrian, and Hung-Yu Kao. "Unsupervised Extractive Summarization-Based Representations for Accurate and Explainable Collaborative Filtering.". In ACL, 2021.
> > >
> > > [8] Zheng, Lei, Vahid Noroozi, and Philip S. Yu. "Joint deep modeling of users and items using reviews for recommendation." Proceedings of the tenth ACM international conference on web search and data mining. 2017.
> > >
> > > [9] Wang, Cheng, Mathias Niepert, and Hui Li. "Recsys-dan: discriminative adversarial networks for cross-domain recommender systems." IEEE transactions on neural networks and learning systems 31.8 (2019): 2731-2740.
> > >
> > > [10] Yu, Wenhui, et al. "Semi-supervised collaborative filtering by text-enhanced domain adaptation." Proceedings of the 26th ACM SIGKDD International Conference on Knowledge Discovery & Data Mining. 2020.

---

> ### Author Response · Authors · 2021-08-26
> **Response to Reviewer RKBg**
>
> Thanks again for the valuable comments. Please let us know if anything is unclear. We truly appreciate this opportunity to improve our work and shall be grateful for any feedback you could give to us.

---

### Official Review · Reviewer_Dtrp · 2021-07-16

**Rating:** 7
**Confidence:** 4

**Summary:**

In this paper, the authors study a relatively new problem called cross-domain cold-start recommendation (CDCSR), where the goal is to transfer knowledge from a warm source domain with user-item ratings and item descriptions to a cold target domain with item descriptions only (i.e., without any user-item interactions). Notice that the users in two domains are aligned. In particular, the authors propose a novel recommendation framework, i.e., DisAlign, which contains two main modules, i.e., a rating prediction module and an embedding distribution alignment module.

**Limitations And Societal Impact:**

Yes

**Main Review:**

1 Originality: The studied problem, i.e., cross-domain cold-start recommendation (CDCSR), is interesting and challenging. The proposed method, i.e., DisAlign, is new since no previous work addresses the CDCSR problem by aligning the latent distributions using Stein path alignment or its improvement version proxy Stein path.

2 Quality: The proposed model is technically sound, which is also supported by the empirical studies.

3 Clarity: Overall, the paper is well presented and is easy to follow. The problem is clearly defined and the techniques are presented with sufficient details (some are in the appendix). Moreover, some illustraitons are included for better understanding, e.g., Figure 1 for the studied problem, Figure 2 fort the motivation of distribution alignment and Figure 3 of the proxy Stein path alignment.

4 Significance:
The authors have conducted extensive empirical studies, including 8 cross-domain settings and 6 baseline methods, as well as ablation studies and case studies.



**Time Spent Reviewing:**

2

---

> ### Author Response · Authors · 2021-08-10
> **Response to Review Dtrp**
>
> Thanks for your careful and valuable comments. We will further improve the paper continuously.

---

### Official Review · Reviewer_NZWF · 2021-07-26

**Rating:** 5
**Confidence:** 4

**Summary:**

Summary:

The paper addresses the cold-start item recommendation with auxiliary information. To attack the heterogeneity/ domain discrepancy across the source domain and target domain, Stein path is adopted to align the domain distributions. In specific, the distribution of target item auxiliary embedding is to be aligned with both source item auxiliary embedding and source item collaborative embedding. A proxy Stein path is proposed to reduce the cubic time complexity. Good experiments are achieved on real-world datasets by comparing with many SOTA methods.


**Main Review:**

Strong points:

S1: Achieving good experimental results, outperforming many SOTA methods.


Weak points:

W1: The technical contribution is limited. It might be novel in applying Stein path to the cold-start item recommendation with auxiliary information.

W2: The motivation is not strong. What is the physical meaning of feeding cold item auxiliary embedding   C with warm item collaborative preference V and auxiliary embedding W to the Stein path?


Other comments:

C1: The examples in Figure 1 and Figure 2 show that the auxiliary information seems to be Visual while it is Text in evaluation experiments.

C2: In terms of Fig 1 and Fig 3, the source domain is Movie while the target domain is Book, while it is not consistent with the body text in Line 138.

C3: in Fig 2, the L_BM should correspond to the L2 Constraint while the L_BK to Item Clustering.


**Time Spent Reviewing:**

6

---

> ### Author Response · Authors · 2021-08-10
> **Response to Review NZWF**
>
> Thanks for your thoughtful comments.  We hope to address your concerns through the following reply.
>
> **Q1**: How about the technical contribution in this paper?
>
> **A1**: First, we propose a general framework (i.e., DisAlign) to solve the CDCSR problem, which consists of the rating prediction module and embedding distribution alignment module. The proposed model can better align the warm and cold item embeddings to provide better recommendation results across domains. We also propose Stein path alignment by calculating the distance between the original target sample to the Stein mirror point. As far as we know, this is the first attempt in literature to adopt Stein path for reducing the domain discrepancy across domains. It is also the first application of SVGD-based method for solving the CDCSR tasks, to our best knowledge. We further propose proxy Stein path alignment with multiple-proxies algorithm in order to reduce the time consumption. We conduct extensive experiments, which shows that our proposed DisAlign model achieves state-of-the-art performance in CDCSR problem. Hopefully, our proposal could shed light on the further development of this research area.
>
> **Q2**: What is the physical meaning of the Stein Path in DisAlign?
>
> **A2**: The Stein path means the Euclidean distance between the original sample points to the Stein mirror points. In our CDCSR scenario, it can be seen as the reproduction of an item from a source domain (e.g., movie domain) to a target domain (e.g., book domain). For example, the stein path of the cold book domain to the warm movie domain, i.e., from $C$ to $V (W)$, can be viewed as the movie reproduction according to the corresponding book contents. Notably that $V$ and $W$ have L2 constraint according to the cold-start problem, and thus aligning the stein path on $C \rightarrow V$ and $C \rightarrow W$ can be helpful in the model training. We will add these descriptions in the final version.
>
> **Q3**: Some typos and other comments.
>
> **A3**: Thanks for pointing out these typos (e.g., line 138 and $L_{BM}$/$L_{BK}$ in Fig.2) and we will fix them in the final version. Besides, as for the auxiliary information in Figure 1 and Figure 2, we show it with visual information rather than text information, because we want to enhance readability and make it clear. In our experiments, we choose text as auxiliary information following existing work [1,2]. In fact, one can choose any auxiliary information when using our proposed model.
>
> **Ref** :
>
> [1] Feng Zhu, Yan Wang, Chaochao Chen, Guanfeng Liu, and Xiaolin Zheng. A graphical and attentional framework for dual-target cross-domain recommendation. In IJCAI, pages 3001–3008, 2020.
>
> [2] Zhihong Chen, Rong Xiao, Chenliang Li, Gangfeng Ye, Haochuan Sun, and Hongbo Deng. Esam: Discriminative domain adaptation with non-displayed items to improve long-tail performance. In SIGIR, pages 579–588, 2020.

---

> ### Author Response · Authors · 2021-08-26
> **Response to Reviewer NZWF**
>
> Thanks again for the valuable comments. Please let us know if anything is unclear. We truly appreciate this opportunity to improve our work and shall be grateful for any feedback you could give to us.

---

### Official Review · Reviewer_6Rcq · 2021-07-29

**Rating:** 9
**Confidence:** 4

**Summary:**

This paper focuses on solving the cold-start issue in Cross-Domain Recommendation (CDR) setting. For this purpose, the authors propose a DisAlign framework, which mainly has two modules, i.e., rating prediction module and embedding distribution alignment module. The main novelty relies in the embedding distribution alignment module. In it, the authors propose Stein path alignment and proxy Stein path alignment. The experiments on benchmark datasets, compared with six baselines, show the superior of the proposed model. The experimental analysis also shows the effectiveness of the model ability.

**Limitations And Societal Impact:**

1. This paper focuses on the setting where all the items are cold in the target domain, but most recommender systems consider some items as warm items and the remaining as cold items. Is this setting also suitable for the proposed method?
2. What specific data pre-processing methods are used during experiments?
3. Why DisAlign-SP outperforms DisAlign-PSP in some cases?


**Main Review:**

- Originality:
Data sparsity is a long-standing problem in recommender systems. Solving this problem in CDR setting by proposing distribution alignment is novel to me. The proposed Stein path alignment and proxy Stein path alignment are novel to me, and the accuracy improvement brought by the proposed models is quite promising.

- Quality:
  This work is technically written with detailed algorithms. Most of the arguments presented in this work are supported by experiment results.

- Clarity:
  This paper is well-organized, and it clearly states the research problem in the introduction. The proposed model is well-motivated and clearly presented.

- Significance:
  To my knowledge, this paper is the first work solving the cold-start problem in CDR setting. The experiments show remarkable improvements (in terms of HR, Recall, and NDCG) over the prior models.


**Time Spent Reviewing:**

7 hours

---

> ### Author Response · Authors · 2021-08-10
> **Response to Review 6Rcq**
>
> Thanks for your careful and valuable comments. We will further improve the paper continuously.
>
> **Q1**: Can some other cases be suitable for the proposed method?
>
> **A1**: Yes, our proposed model can be easily modified to solve the problem when some of the target items have the rating interactions, and we provide the solution as below. First, for the cold items in the target domain, we adopt Stein path alignment or proxies Stein path alignment for aligning the auxiliary embeddings between the target warm items and source items, the same as our current solution. Second, for the rest warm items in the target domain, we directly adopt the rating prediction module to model the item embeddings and auxiliary embeddings. We will leave the details as a future work.
>
> **Q2**: How about the specific data pre-processing methods are used during experiments?
>
> **A2**: We mainly adopt the data pre-processing methods to deal with the auxiliary representations. Specifically, we first collect the tags, profiles, and item details for each item in both source and target domains. After that, we filter the stopping words and punctuations, and adopt Word2Vec [1,2] to obtain each word's embedding. Then, we average the embeddings of features above to obtain the initial auxiliary representation for each item. We simply describe these procedures in line 267-269.
>
> **Q3**: Why DisAlign-SP outperforms DisAlign-PSP in some cases?
>
> **A3**: Based on the empirical study, we found that DisAlign-SP outperforms DisAlign-PSP when the cold-item embeddings space is scattered, since it will be difficult to figure out the typical samples. Under such situations, the proxies cannot fully illustrate the whole cold-item embedding space, which will finally degrade the model performance.The result demonstrates that the proxy stein path alignment is more suitable when the embedding space is clustered, since it can filter out the outliers to provide better performance.
>
> **Ref** :
>
> [1] Yan Song, Shuming Shi, Jing Li, and Haisong Zhang. Directional skip-gram: Explicitly distinguishing left and right context for word embeddings. In Proceedings of the 2018 Conference of the North American Chapter of the Association for Computational Linguistics: Human Language Technologies, Volume 2 (Short Papers), pages 175-180, 2018.
>
> [2] Jeffrey Pennington, Richard Socher, and Christopher Manning. Glove: Global vectors for word representation. volume 14, pages 1532-1543, 01 2014.

---

### Author Response · Authors · 2021-09-01
**Thanks to All The Reviewers**

We would like to thank all the reviewers for your detailed and valuable comments, and particularly, for recognizing the potential of our work in solving the Cross-Domain Cold-Start Recommendation (CDCSR) problem.

We hope our response has adequately addressed the main concerns regarding the motivation, technical limitation, data preprocessing in experiments, and application scenarios. Based on the discussions with reviewers, by far they have verified the importance of the CDCSR problem (by reviewers **6Rcq** and **Dtrp**), the novelty of our technical solution based on Stein path (by reviewers **Dtrp**, **RKBg**, and **6Rcq**), and the significances of the experiments (by reviewers **6Rcq**, **NZWF**, **Dtrp**, and **RKBg**).

Kindly let us know if anything is unclear. We truly appreciate your valuable feedback and comments that help us further improve our work.

---

### Decision · Program_Chairs · 2021-09-27

**Decision:**

Accept (Poster)

**Comment:**

Reviews are quite mixed, and admittedly the one most positive review is somewhat of an outlier, though there's enough here to recommend acceptance. Still somewhat borderline given the two marginal scores (2x5), though with two strong voices for acceptance I'd lean positive.

In spite of the mixed scores, and the detailed rebuttal, opinions didn't really move during the discussion phase. So mostly my judgement is based on the initial scores, as well as what appears to be a reasonable rebuttal (even if it didn't move the needle on scores).

Reviewers were not totally aligned in their opinions. Some praised the originality and significance of the paper (especially the most positive review), while others found the technical contribution more limited. Other issues centered around lack of clarity w.r.t. various components. Ultimately the negative points seemed not to be deal-breakers and could be fixed easily.

Again, enough here to recommend acceptance, and nothing that seems like a red flag, though still somewhat borderline in terms of scores / distribution of scores.